# DYNAMIC HISTORICAL ADAPTATION FOR CONTINUAL IMAGE-TEXT MODELING

## ABSTRACT

In realistic application scenarios, existing methods for image-text modeling have limitations in dealing with data stream: training on all data needs too much computation/storage resources, and even the full access to previous data is invalid. In this work, we thus propose a new continual image-text modeling (CITM) setting that requires a model to be trained sequentially on a number of diverse image-text datasets. Although recent continual learning methods can be directly applied to the CITM setting, most of them only consider reusing part of previous data or aligning the output distributions of previous and new models, which is a partial or indirect way to acquire the old knowledge. In contrast, we propose a novel dynamic historical adaptation (DHA) method which can holistically and directly review the old knowledge from a historical model. Concretely, the historical model transfers its total parameters to the main/current model to utilize the holistic old knowledge. In turn, the main model dynamically transfers its parameters to the historical model at every five training steps to ensure that the knowledge gap between them is not too large. Extensive experiments show that our DHA outperforms other representative/latest continual learning methods under the CITM setting.

## 1 INTRODUCTION

In the past few years, image-text modeling has drawn much attention from both academia and industry with a fundamental role in various cross-modal tasks, such as image-text retrieval (Chen et al., 2020a; Lee et al., 2018), image captioning (Vinyals et al., 2015; Jia et al., 2015), and text-image generation (Johnson et al., 2018; Qiao et al., 2019). Although existing image-text modeling methods (Lu et al., 2019; Li et al., 2020; Lei et al., 2021; Yang et al., 2021; Ging et al., 2020; Bain et al., 2021; Huo et al., 2021; Jia et al., 2021) have achieved great success in these tasks, most of them assume that a full (fixed) set of image-text pairs are provided for model training, which actually limits their deployment in realistic application scenarios. That is, the training data often comes in a stream way, and the current widely-used paradigm for image-text modeling faces two limitations: (1) training on all data (i.e., both previous and new data) severely increases the computational and storage overhead; (2) the full access to previous data may be invalid.

To overcome these limitations, we thus propose a continual image-text modeling (CITM) setting instead. Concretely, we recollect four diverse image-text datasets respectively from MSCOCO (Lin et al., 2014), CC3M (Sharma et al., 2018), WIT (Srinivasan et al., 2021) and GoodNews (Biten et al., 2019), each of which is split into the training, validation, and test sets. We adopt the SimCLR-based model (Chen et al., 2020b) as the basic model which is also deployed in OpenAI CLIP (Radford et al., 2021). Under the CITM setting, the model is sequentially trained on each of the four image-text datasets, and is finally evaluated on all datasets. To demonstrate the well-known catastrophic forgetting problem, we measure the image-to-text retrieval

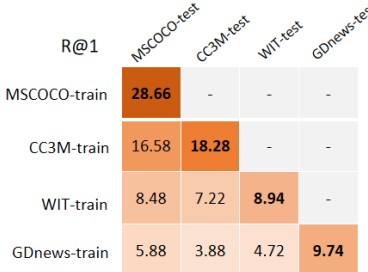

Figure 1: The results of catastrophic forgetting under the CITM setting.

performance with the metric recall@1 (R@1) during sequential training on the four datasets. The results in Figure 1 clearly show that every time the model is trained on a new dataset, its performance on previous datasets has a distinct degradation (i.e., catastrophic forgetting).

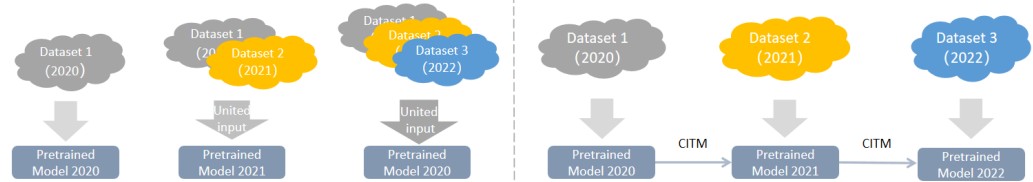

Figure 2: Schematic illustration of the realistic application of our proposed CITM setting in large-scale multi-modal pre-training (like OpenAI CLIP) with the pre-training data being updated every year. *Left:* The traditional setting for large-scale pre-training with annual data update. *Right:* Our CITM setting for large-scale pre-training with annual data update.

Among existing continual learning methods, rehearsal-based (Chaudhry et al., 2019; Buzzega et al., 2020) and regularization-based methods (Li & Hoiem, 2017; Rannen et al., 2017; Zhang et al., 2020; Cha et al., 2021) can be easily applied to the CITM setting, while architecture-related methods (Mallya & Lazebnik, 2018; Mallya et al., 2018; Rosenfeld & Tsotsos, 2018) generally need extra task-specific modules and are unsuitable for CITM with a unified architecture. In this paper, we thus devise baseline methods for CITM mainly by deploying rehearsal-based and regularization-based methods. Note that these two groups of continual learning methods have their own limitations. Specifically, rehearsal-based methods set up a memory buffer to replay previous data, and only preserve partial old knowledge due to the sample selection imposed on the memory buffer. Moreover, regularized-based methods can only convey the old knowledge by aligning the output distributions of the previous and new models, which indicates that the old knowledge from the previous model can only be indirectly transferred through data-driven guidance. Such an indirect approach is thus vulnerable to large domain shifts across the previous and new tasks.

To avoid the drawbacks of the above baseline methods for CITM, we thus propose a novel dynamic historical adaptation (DHA) method which can holistically and directly review the old knowledge from a historical model. The core idea of our DHA is to directly transfer knowledge between the old and new models through parameter interaction. In our DHA, we name the model trained on the current task as the main model, and the best (main) model on the last task as the historical model. During parameter interaction, we directly transfer the parameters of the historical model to the main model and then train the main model with modified parameters on the current task. Meanwhile, we dynamically update the historical model with the guidance of the main model to ensure that the knowledge gap between them is not too large. Specifically, at every five steps, the parameters of the main model are passed to the historical model for parameter modification. Overall, these two parameter transfer strategies make up our DHA method. Compared with existing methods (Li & Hoiem, 2017; Chaudhry et al., 2019; Buzzega et al., 2020; Cha et al., 2021), our DHA has two advantages: (1) DHA adopts direct parameter transfer instead of indirect model aligning (deployed by regularization-based methods), and thus it is more robust to large domain shifts across the previous and new tasks. (2) DHA holistically reviews the old knowledge from the historical model, which can overcome the drawback of rehearsal-based methods for partial data selection (i.e., partial old knowledge is reused). To our best knowledge, we are the first to propose a direct parameter transfer method to cope with the forgetting problem in the continual learning field.

As we have mentioned, we construct a benchmark dataset for the CITM setting by recollecting four diverse image-text datasets respectively from MSCOCO (Lin et al., 2014), CC3M (Sharma et al., 2018), WIT (Srinivasan et al., 2021) and GoodNews (Biten et al., 2019). Under a fair setting, we compare DHA with a number of baseline methods (Li & Hoiem, 2017; Chaudhry et al., 2019; Buzzega et al., 2020; Cha et al., 2021) on this benchmark dataset. Extensive experiments prove that our DHA outperforms these baseline methods under the CITM setting.

Overall, the main contributions of this paper can be summarized as follows: (1) We propose a new continual image-text modeling (CITM) setting for image-text modeling on data stream, which has a realistic application in large-scale multi-modal pre-training (with annual data update) as shown in Figure 2. (2) We devise a novel dynamic historical adaptation (DHA) method under the CITM setting. For the first time, we identify the important role of direct parameter transfer (between the historical and main models) in continual learning. (3) We construct a benchmark dataset of four diverse sets of image-text pairs, which can facilitate the research on CITM. (4) Extensive experiments demonstrate the effectiveness of our DHA under the CITM setting.

## 2 RELATED WORK

**Image-Text Modeling.** Recent image-text modeling methods can be summarized into two groups: single-stream and two-stream methods. **(1)** Single-stream methods aim to learn the unified representation of the image-text pair with a fusion module. Most of existing single-stream methods (Lu et al., 2019; Tan & Bansal, 2019; Zhu & Yang, 2020; Li et al., 2020; Lei et al., 2021; Yang et al., 2021) choose to concatenate the image and text embeddings as the input of the fusion module (e.g., cross-attention transformer). Although model training is easy for single-stream methods, it requires calculating the similarities of all the possible query-candidate pairs during inference. Therefore, they suffer from heavy computation burdens. **(2)** Two-stream methods (Ging et al., 2020; Patrick et al., 2020; Bain et al., 2021; Huo et al., 2021; Jia et al., 2021; Radford et al., 2021) adopt independent image and text encoders to learn image and text embeddings that are aligned in a joint space. Compared to single-stream methods, two-stream methods allow different depths and designs of network architectures for the two modalities and enjoy much more efficient inference. In this work, we follow the two-stream architecture for image-text modeling: ResNet50 (He et al., 2016) is used as the image encoder, and BERT-base (Devlin et al., 2018) is used as the text encoder. We adopt SimCLR (Chen et al., 2020b) as the basic contrastive learning method for model training.

**Continual Learning.** By reviewing recent progress in conventional continual learning, we can divide main-stream approaches into three groups: **(1) Rehearsal-Based Methods.** Early classic rehearsal-based method (Rebuffi et al., 2017) proposes to store part of exemplars of previous classes in order to acquire better class means. (Chaudhry et al., 2019) finds that retraining a subset of old data on new tasks can help address the forgetting problem and also provides several memory update strategies. (Aljundi et al., 2019; Chaudhry et al., 2021; Buzzega et al., 2020) further explore the approaches to selecting representative samples from old tasks. In addition, pseudo-data rehearsal generating approaches (Shin et al., 2017; Atkinson et al., 2018; Lavda et al., 2018; Liu et al., 2020; Ramapuram et al., 2020) are proposed to avoid extra storage and generate more representative samples for training, whereas generating pseudo-data actually increases the training time. Note that the rehearsal-based methods suffer from the drawback that only partial historical knowledge is transferred by the memory buffer. **(2) Regularization-Based Methods.** This group of methods mainly aim to distill the knowledge of the previous models. (Li & Hoiem, 2017; Rannen et al., 2017; Zhang et al., 2020) align the output features or logits between the previous and the current models with an extra regularization penalty. Since the domain shifts exist across different tasks, such regularization penalty brings additional training difficulty (De Lange et al., 2021). Other methods (Aljundi et al., 2018; Chaudhry et al., 2018; Kirkpatrick et al., 2017) constrain part of the parameters of the model. Since most of these methods are designed for classification tasks, they are hard to be directly applied to our CITM setting. **(3) Architecture-Related Methods.** This group of methods mitigate the difference in new tasks in two ways. (Mallya & Lazebnik, 2018; Mallya et al., 2018; Serra et al., 2018) mask different parameters while training different tasks. (Aljundi et al., 2017; Rosenfeld & Tsotsos, 2018; Xu & Zhu, 2018) extend network architecture for new tasks. A potential drawback of these methods is that they generally need extra task-specific modules and are unsuitable for CITM with a unified architecture. Other than the above approaches with a single strategy, recent works (Buzzega et al., 2020; Cha et al., 2021) start to design combined strategies for continual learning based on rehearsal-based and regularization-based methods. Finally, we notice that most of existing continual learning approaches have a common characteristic that the old knowledge is expressed with (partial) data, which means that the model update to mitigate forgetting may be affected by partial/indirect guidance. In contrast, our proposed DHA provides a new perspective of continual learning that the old knowledge could be holistically preserved by direct parameter transfer.

## 3 PROPOSED METHOD

### 3.1 PRELIMINARY

We first define our proposed CITM setting formally. Given a sequence of $n$ image-text datasets $\mathcal{D} = \{D_1, D_2, ..., D_n\}$ coming from $n$ domain sources like a stream, a model for CITM is supposed to be sequentially trained on $\mathcal{D}$. Each dataset $D_t$ ($1 \leq t \leq n$) is defined as $D_t = \{(x_i^I, x_i^T)\}_{i=1}^{N_t}$, where $x_i^I$ and $x_i^T$ respectively denote the image and text samples in the $i$-th image-text pair, and $N_t$ denotes the number of data pairs. The image-text retrieval task (Chen et al., 2020a; Lee et al., 2018) on each dataset $D_t$ is denoted as $T_t$ ($1 \leq t \leq n$). For each task $T_t$, a model for image-text

retrieval typically learns to align the image and text embeddings with contrastive loss Chen et al. (2020b). Under the CITM setting, the model only concentrates on the current task during sequential training, leading to the catastrophic forgetting of previous knowledge. For performance evaluation, the obtained final model (trained across all tasks) is tested on each of the $n$ tasks.

## 3.2 NETWORK ARCHITECTURE

Under the CITM setting, we propose a novel dynamic historical adaptation (DHA) method which can holistically and directly review the old knowledge from a historical model. The core idea of our DHA is to directly transfer knowledge between the old and new models through parameter interaction. To this end, our DHA model is devised to have two key components: the historical model and the main model, as illustrated in Figure 3. These two models share the same architecture while only the main model requires the backward update. We follow the two-stream network architecture like CLIP (Radford et al., 2021), which has achieved remarkable performance in image-text retrieval tasks. Concretely, the image encoder takes ResNet50 as the backbone and the text encoder takes BERT-Base as the backbone, which are both initialized with unimodal pre-trained models.

**Image and Text Encoders.** Formally, the backbone ResNet50 of the image encoder is denoted as $f_{ResNet}^I$. Meanwhile, the backbone BERT-Base of the text encoder is denoted as $f_{Bert}^T$. Given an input text $x_i^T$, we first tokenize it into a sequence as $[tk_i^1, tk_i^2, ..., tk_i^{l_i}]$, where $l_i$ denotes the length of $x_i^T$. To ensure that the text and image embeddings have the same dimension, we append linear projection layers $f_P^I$ and $f_P^T$ to ResNet50 and BERT-Base, respectively. Given an image-text pair $(x_i^I, x_i^T)$, the final image and text embeddings are given by:

$$e_i^I = f_P^I(f_{ResNet}^I(x_i^I)), \tag{1}$$

$$e_i^T = f_P^T(f_{Bert}^T(tk_i^1, tk_i^2, ..., tk_i^{l_i})). \tag{2}$$

**Contrastive Loss Function.** Since our proposed DHA has the two-stream architecture, it can be effectively trained by the well-known contrastive learning method SimCLR (Chen et al., 2020b). Concretely, given a batch of $B$ image-text pairs $\{x_i^I, x_i^T\}_{i=1}^B$ during training, the loss function is constructed as follows. For each input image $x_i^I$, we define the contrastive loss between its image embedding $e_i^I$ and the embeddings of all positive/negative texts in the batch as an InfoNCE loss:

$$L_c^{i2t} = -\frac{1}{B} \sum_{i=1}^B \log \frac{exp(e_i^I \cdot e_i^T / \tau)}{exp(e_i^I \cdot e_i^T / \tau) + \sum_{j \neq i} exp(e_i^I \cdot e_j^T / \tau)}, \tag{3}$$

where $\tau$ denotes the temperature hyperparameter, and the vector similarity is measured by dot product ($\cdot$). Similarly, for each input text $x_i^T$, the InfoNCE loss is given by:

$$L_c^{t2i} = -\frac{1}{B} \sum_{i=1}^B \log \frac{exp(e_i^I \cdot e_i^T / \tau)}{exp(e_i^I \cdot e_i^T / \tau) + \sum_{j \neq i} exp(e_j^I \cdot e_i^T / \tau)}. \tag{4}$$

The total contrastive loss for training our DHA is thus defined as:

$$L_c = L_c^{i2t} + L_c^{t2i}. \tag{5}$$

In this work, for fair comparison, all the competitors for CITM adopt the same network architecture and the same basic contrastive loss function as our DHA. More details can be found in Sec. 4.

## 3.3 DYNAMIC HISTORICAL ADAPTATION

As we have mentioned, our motivation of method design is to transfer the holistic knowledge contained in the historical model to the new model, without suffering from the drawbacks of existing continual learning methods. Concretely, data rehearsal approaches (Chaudhry et al., 2019; Buzzega et al., 2020) attempt to preserve the previous data distribution, but retaining a memory buffer of limited size may cause the overfitting to the partial samples of the previous task. Moreover, regularization-based approaches (Li & Hoiem, 2017; Rannen et al., 2017; Zhang et al., 2020; Cha et al., 2021) attempt to store the historical knowledge by aligning the historical and main models with regularization-based penalty terms, but such an indirect way to addressing the forgetting

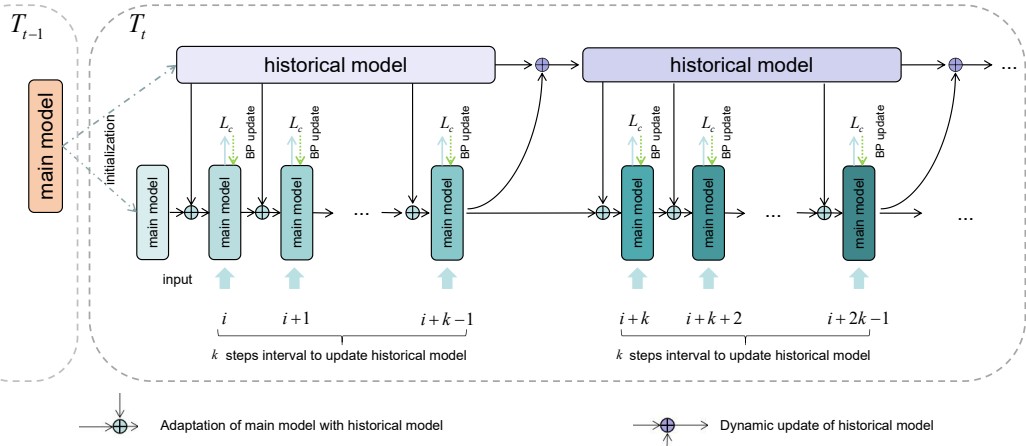

Figure 3: Overview of the proposed DHA method for the CITM setting. At the beginning of task $T_t$, the best main model found in task $T_{t-1}$ (according to the validation performance on task $T_{t-1}$) is used to initialize both the historical and main models. During each training iteration, the main model first receives the transferred parameters from the historical model and then learns on the dataset of task $T_t$. Moreover, for every $k$ training iterations, the historical model is updated with the transferred parameters of the main model.

problem is thus vulnerable to large domain shifts across the previous and new tasks. In this paper, we thus propose a novel dynamic historical adaptation (DHA) method which can holistically and directly review the old knowledge from a historical model. Below we introduce the details of the two update strategies applied in our DHA throughout training.

**Adaptation of Main Model with Historical Model:** Since all the learned knowledge has been held and expressed by model parameters, we believe that directly transferring the parameters of the historical model to the main model is a direct and effective approach to preserving the historical knowledge. The direct parameter transfer process is shown in Figure 3. Formally, let $\theta_H$, $\theta_M$, $\theta_M^*$ denote the parameters of the historical model, the main model, and the best main model found in the last task, respectively. Moreover, let $\theta_H^i$ and $\theta_M^i$ denote the parameters of the historical model and the main model at the end of the $i$-th training iteration in the current task, respectively. At the beginning of the current task, we initialize the main model and the historical model with the parameters of the best main model found in the last task (i.e., $\theta_M^0 = \theta_M^*$ and $\theta_H^0 = \theta_M^*$). For each training iteration ($i \geq 1$) before data load, we choose to update $\theta_M^{i-1}$ with part of $\theta_H^{i-1}$ and obtain $\theta_M^{i_{pre}}$ as the new intermediate parameters of the main model. After such parameter update, the main model is trained on the input data and backward updated normally to obtain $\theta_M^i$ as the final parameters of the $i$-th training iteration. We define the gradient function w.r.t. $\theta_M$ as:

$$G_{L_c}(\hat{\theta}_M) = \left. \frac{\partial L_c}{\partial \theta_M} \right|_{\theta_M = \hat{\theta}_M}. \tag{6}$$

The above adaptation strategy for the main model with the historical model can be formulated as:

$$\theta_M^{i_{pre}} = \lambda_1 \theta_M^{i-1} + (1 - \lambda_1)\theta_H^{i-1}, \tag{7}$$

$$\theta_M^i = \theta_M^{i_{pre}} - \eta G_{L_c}(\theta_M^{i_{pre}}), \tag{8}$$

where $\eta$ denotes the learning rate, and $\lambda_1$ denotes the weighting coefficient. By combining Eq. (7) and Eq. (8), we have the adaptation process from $\theta_M^{i-1}$ to $\theta_M^i$ as follows:

$$\theta_M^i = \lambda_1 \theta_M^{i-1} + (1 - \lambda_1)\theta_H^{i-1} - \eta G_{L_c}(\lambda_1 \theta_M^{i-1} + (1 - \lambda_1)\theta_H^{i-1}). \tag{9}$$

**Dynamic Update of Historical Model:** Currently, the main model has received the guidance from the historical model. However, since the parameters of the historical model remain static in the current task, this may cause two concerns: (1) Since the main model always learns better on the current task as the training process goes on, the knowledge gap between the historical and main

models is gradually enlarged. Therefore, the parameters transferred from the unchanged historical model tend to cause degradation to the retrieval performance of the main model on the current task. (2) Such performance degradation to the main model on the current task would finally affect the performance of the final model (i.e. the best main model across all tasks) when it is evaluated on this task. To address these concerns, we choose to make the parameters of the historical model gradually change by updating it with the parameters of the main model (but not so frequently) . This dynamic update of the historical model at the $i$-th training iteration is given by:

$$\theta_H^i = \begin{cases} \lambda_2 \theta_H^{i-1} + (1 - \lambda_2)\theta_M^{i-1}, & \text{if } i = mk, \, m \in \mathbb{N} \\ \theta_H^{i-1}, & \text{otherwise} \end{cases}, \tag{10}$$

where $k$ denotes the step interval for model update, and $\lambda_2$ denotes the weighting coefficient.

Overall, our proposed DHA is composed of the above two update strategies, which have been shown to be effective in Sec. 4.3 and Sec. 4.4. We believe that direct parameter transfer is another promising way to handling the continual learning problem in image-text modeling. The pseudocode of the full algorithm for our proposed DHA is given in Appendix A.

## 4 EXPERIMENTS

### 4.1 EXPERIMENTAL SETUP

**Datasets.** To mimic the realistic application of the CITM setting in multi-modal per-training (like OpenAI CLIP), we recollect four image-text datasets for benchmark construction from the following large diverse datasets of image-text pairs: **(1) MSCOCO (Lin et al., 2014)** is an image-text dataset that consists of $123,287$ images with their captions. Each image is annotated with 5 captions. Most images are related to the nature and common objects in daily life. **(2) CC3M (Sharma et al., 2018)** is a well-known image-captioning dataset for image-text pre-training. It is composed of about $3M$ image-text pairs, which are collected from the Internet with weak relation between images and their textual descriptions. **(3) WIT (Srinivasan et al., 2021)** is a large multimodal multilingual dataset collected from the Wikipedia website. This dataset has a total of $11.5M$ images. Each image is annotated with the corresponding textual description or contextual information. **(4) GoodNews (Biten et al., 2019)** is a large news image-captioning dataset. It is collected from the New York Times. Unlike the other datasets, the captions in GoodNews are written by professional journalists and thus are claimed to have implications for the style and richness of the news. In this paper, based on the aforementioned four image-text datasets, our benchmark dataset of four sequential tasks is constructed as follows: (1) For task $T_1$, we randomly select $100,000$ images with corresponding captions from MSCOCO as the training set, $13,287$ as the validation set, and $5,000$ as the test set. (2) For the other tasks $T_2 - T_4$, we construct the three task-specific datasets from CC3M, WIT, and GoodNews, respectively. Concretely, the training/validation/test set is formed to have $130,000/13,000/5,000$ image-text pairs uniformly for each of $T_2 - T_4$.

**Evaluation Metrics.** We adopt **Recall@mean** (R@mean) and **Forgetting Rate** (FR) as our evaluation metrics. **R@mean** indicates the mean value of Recall@1, Recall@5, and Recall@10, where Recall@K (K=1,5,10) denotes the percentage of correct matching in the top-K retrieved results. The R@mean on each task indicates the retrieval performance of the final model on this task. Moreover, for the final model tested on task $T_t$, **FR** is defined as $\text{FR}_t^n = \frac{R_t^t - R_t^n}{R_t^t}$, where $R_t^t$ denotes the R@mean of the best main model in task $T_t$ on the test set of task $T_t$, and $R_t^n$ denotes the performance of the final model on the test set of task $T_t$. The average FR is $\text{FR} = \frac{1}{n-1}\sum_{t=1}^{n-1}\frac{R_t^t - R_t^n}{R_t^t}$.

### 4.2 IMPLEMENTATION DETAILS

Under the CITM setting, we train our DHA model on a sequence of four datasets: MSCOCO (Task $T_1$), CC3M (Task $T_2$), WIT (Task $T_3$), and GoodNews (Task $T_4$). After the main model has completed its training on task $T_{t-1}$, we find the best main model on the validation set of $T_{t-1}$. At the beginning of task $T_t$, this best main model is used to initialize both the main and history models on this new task. For fair comparison, we set the memory buffer to have $5\%$ samples of the train set of each task for our DHA (if buffer is used) and all competitors. The details of the memory buffer updating strategy are included in Appendix A. To make comprehensive study, we implement our DHA with and without memory buffer to validate its effectiveness under the CITM setting.

Table 1: Comparative results between our DHA and other representative/latest methods. 'T2I' denotes text-to-image retrieval and 'I2T' denotes image-to-text retrieval. All methods adopt the same network architecture. 'Mem' denotes the data rehearsal with 5% buffer.

| | Method | Mem | Task $T_1$ | | Task $T_2$ | | Task $T_3$ | | Task $T_4$ | Average | |
|---|---|---|---|---|---|---|---|---|---|---|---|
| | | | R@mean | FR | R@mean | FR | R@mean | FR | R@mean | R@mean | FR |
| T2I | Baseline | N | 13.64 | 68.30 | 12.78 | 64.82 | 13.63 | 40.25 | 22.62 | 15.67 | 57.79 |
| | LwF (Li & Hoiem, 2017) | N | 16.81 | 60.94 | 15.69 | 56.31 | 15.33 | 31.41 | 22.68 | 17.63 | 49.55 |
| | ER (Chaudhry et al., 2019) | Y | 16.30 | 62.12 | 16.15 | 55.08 | 14.37 | 36.42 | 21.57 | 17.10 | 51.21 |
| | DER (Buzzega et al., 2020) | Y | 20.52 | 52.31 | 20.92 | 41.74 | **16.31** | **24.07** | 21.04 | 19.70 | 39.37 |
| | CO2L (Cha et al., 2021) | Y | 19.64 | 54.35 | 18.95 | 46.14 | 16.13 | 24.94 | **22.95** | 19.42 | 41.81 |
| | DHA$^\dagger$ (ours) | N | 21.31 | 50.48 | 21.82 | 37.15 | 15.64 | 27.09 | 21.37 | 20.04 | 38.24 |
| | DHA (ours) | Y | **24.58** | **42.88** | **22.95** | **33.82** | 16.15 | 24.50 | 21.22 | **21.29** | **33.73** |
| I2T | Baseline | N | 17.26 | 66.48 | 11.55 | 68.60 | 13.48 | 42.02 | **23.72** | 16.50 | 59.03 |
| | LwF (Li & Hoiem, 2017) | N | 21.59 | 58.07 | 15.36 | 57.58 | 15.42 | 34.69 | 23.29 | 18.92 | 50.11 |
| | ER (Chaudhry et al., 2019) | Y | 21.23 | 58.57 | 15.17 | 57.79 | 14.79 | 37.83 | 22.03 | 18.31 | 51.40 |
| | DER (Buzzega et al., 2020) | Y | 27.55 | 46.49 | 19.08 | 47.31 | 16.34 | 26.43 | 21.99 | 21.24 | 40.08 |
| | CO2L (Cha et al., 2021) | Y | 26.23 | 49.06 | 17.09 | 52.18 | 16.33 | 27.67 | 23.59 | 20.81 | 42.97 |
| | DHA$^\dagger$ (ours) | N | 27.91 | 45.80 | 17.69 | 50.28 | 15.60 | 30.11 | 22.02 | 20.81 | 42.06 |
| | DHA (ours) | Y | **32.72** | **36.45** | **21.01** | **39.50** | **16.45** | **24.71** | 22.25 | **23.11** | **33.55** |

We adopt BERT-Base (Devlin et al., 2018)/ResNet50 (He et al., 2016) as the backbone of text/image encoder. They both use corresponding unimodal pre-trained models for initialization. The images are resized to 224x224 pixels, and the max length of the text descriptions is set to 256 (tokens). We set the learning rate at the beginning of each task to 5e-5 and multiply it by 0.1 as the validation loss does not decrease. We adopt the optimizer Adam for gradient propagation, with the weight decay $1e^{-5}$. The batch size is set to 320 for each training iteration. $\lambda_1$, $\lambda_2$, and $k$ are empirically selected as 0.995, 0.985 and 5, respectively. The main model is trained for 15 epochs on the training set of each task. The total training time on four datasets is around 12 hours with 8 Tesla V100 GPUs. The dataset and code will be released soon.

## 4.3 MAIN RESULTS

We compare our DHA with other representative/latest methods, including the classic regularized-based method LwF (Li & Hoiem, 2017), the classic rehearsal-based method ER (Chaudhry et al., 2019), and two fusion methods DER (Buzzega et al., 2020) and CO2L (Cha et al., 2021) which combine the regularized-based and rehearsal-based strategies (the implementation details of these competitors are included in Appendix B). The basic method (denoted as 'Baseline') denotes training the same network sequentially on four tasks but without any continual learning strategy. The comparative results in Table 1 (see more results in Appendix D) show that: (1) Our DHA beats all the competitors according to average R@mean and average FR over all tasks. The margins between our DHA and all the competitors are especially significant on average FR. This suggests that our direct parameter transfer strategy used for designing DHA is indeed effective for the CITM setting. (2) Our DHA outperforms the second best method DER by 1.59% – 1.87% on average R@mean and 5.64% – 6.53% on average FR. This further validates the effectiveness of our direct parameter transfer strategy used for designing DHA. (3) Our DHA$^\dagger$ (without memory buffer) achieves better results than most of the other approaches. When the rehearsal-based strategy is fused, our DHA achieves the state-of-the-art results. That is, our DHA provides a new promising approach to continual image-text modeling. (4) On the most previous tasks (e.g., $T_1$ and $T_2$), our DHA performs significantly better than all the competitors in preserving much earlier knowledge. This superior ability would make a greater difference in realistic applications when there are more tasks in the data stream. (5) On the newest task $T_4$, nearly all the methods cause a drop on R@mean as compared to 'Baseline'. Such performance drop is mainly due to the trade-off between preserving previous knowledge and learning the current task, which is a common practice in continual learning scenarios.

To show more detailed performance of all methods in alleviating forgetting, we provide the results of the main model (of all methods) on task $T_1$ during sequential training on the four tasks in Figure 4 (more results on task $T_2$ and task $T_3$ are shown in Appendix D). It actually shows the change tendency of R@mean on task $T_1$ of the main model when it is being trained on the later tasks sequentially. Specifically, the left sub-figure show the text-to-image retrieval performance on task $T_1$ when the main model is trained from task $T_1$ to task $T_4$, while the right sub-figure show the corresponding

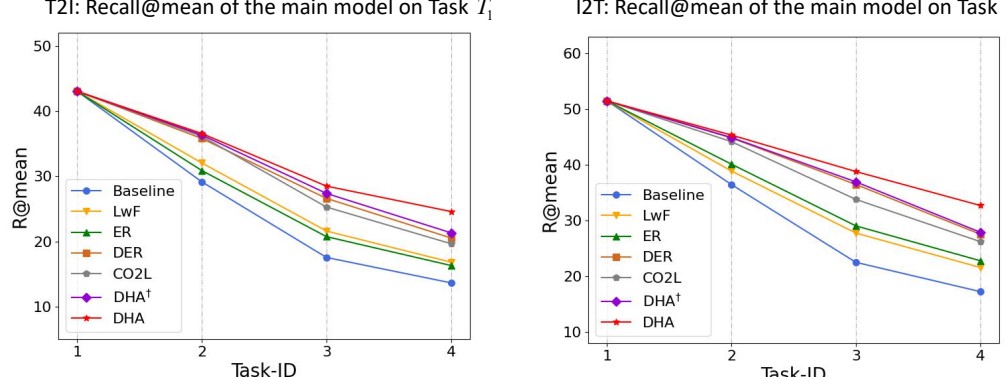

Figure 4: Illustration of the results of the main model (of all methods) on Task $T_1$ during sequential training on the four tasks. 'T2I' denotes text-to-image retrieval and 'I2T' denotes image-to-text retrieval. It can be clearly seen that our DHA forgets with the lowest speed.

Table 2: Direct retrieval results on the test set of Flickr30K obtained by our DHA and other representative methods. Note that all methods are trained on the sequence of four image-text datasets (i.e., {MSCOCO, CC3M, WIT, GoodNews}) under the CITM setting.

| Method | T2I | | | | I2T | | | |
|---|---|---|---|---|---|---|---|---|
| | R@1 | R@5 | R@10 | R@mean | R@1 | R@5 | Recall@10 | R@mean |
| Baseline | 11.04 | 29.70 | 40.42 | 27.05 | 16.30 | 36.60 | 47.00 | 33.30 |
| ER (Chaudhry et al., 2019) | 12.46 | 32.74 | 44.30 | 29.83 | 17.90 | 40.80 | 51.30 | 36.67 |
| DER (Buzzega et al., 2020) | 16.52 | 39.28 | 51.46 | 35.93 | 22.70 | 47.60 | 59.40 | 43.23 |
| DHA† (ours) | 16.02 | 36.58 | 47.52 | 33.52 | 21.50 | 47.10 | 58.80 | 42.47 |
| DHA (ours) | **17.76** | **41.30** | **54.22** | **37.76** | **24.00** | **48.90** | **63.10** | **45.33** |

image-to-text performance. It can be clearly seen that: (1) Our DHA helps the main model forget with the slowest speed during sequential training among all the methods under the CITM setting. (2) Even without memory data, the forgetting speed of our DHA† is still slower than that of most of the other competitors. Overall, these observations provide further evidence that our direct parameter transfer strategy (used in DHA) is indeed effective in alleviating forgetting, and our DHA can be deployed as a new promising approach to continual image-text modeling.

Additionally, we conduct direct retrieval experiments on the test set of Flickr30K (Young et al., 2014), which has no overlap with the sequence of four image-text datasets (i.e., {MSCOCO, CC3M, WIT, GoodNews}) under the CITM setting. We compare our DHA with Baseline, ER, and the best competitor DER, which are all sequentially trained on the four datasets. The comparative results in Table 2 show that our DHA achieves the best performance, i.e., our DHA has the strongest generalization ability due to the direct parameter transfer strategy used for alleviating forgetting.

### 4.4 ABLATION STUDY

Our proposed DHA is composed of two main strategies: (1) adaptation of the main model with the historical model (shortened as 'Adapt with Hist'), i.e., the main model keeps reviewing the historical knowledge by receiving the parameters of the historical model; (2) dynamic update of the historical model (shortened as 'Dynamic Hist'), i.e., the historical model is renewed by updating its parameters with the parameters of the main model. To clearly show the influence of each strategy on the model performance and also study the effect of the memory buffer, we provide the ablation study results for our full DHA on image-to-text retrieval in Table 3. We only show the results (R@mean) of the final model (trained across all four tasks) on each task under the CITM setting. We can observe that: (1) The most basic method with no DHA strategies and no memory buffer has the lowest performance on average R@mean. (2) Only adopting the strategy of 'Adapt with Hist' yields a 3.85% improvement on average R@mean, showing that it can well retain the knowledge of the previous tasks. (3) Adopting both 'Adapt with Hist' and 'Dynamic Hist' strategies brings further improvements on average R@mean. Particularly, such fusion yields performance gains on tasks $T_2$, $T_3$, and $T_4$ out of all the four tasks. This actually validates the effectiveness of 'Dynamic Hist': by controlling the gap between the historical and main models, the found best main model of task

Table 3: Ablation study results (R@mean) for our full DHA under the CITM setting. 'Adapt with Hist' denotes adaptation of the main model with the historical model. 'Dynamic Hist' denotes dynamic update of the historical model. 'Mem' denotes the data rehearsal with 5% buffer. The second best results are highlighted by underline.

| Adapt with Hist | Dynamic Hist | Mem | Task $T_1$ | Task $T_2$ | Task $T_3$ | Task $T_4$ | Average |
|---|---|---|---|---|---|---|---|
| | | | 17.59 | 12.33 | 12.12 | **22.80** | 16.21 |
| ✓ | | | 32.30 | 17.09 | 12.29 | 18.57 | 20.06 |
| ✓ | ✓ | | 27.91 | 17.69 | 15.60 | 22.02 | 20.81 |
| | | ✓ | 21.23 | 15.17 | 14.79 | 22.03 | 18.31 |
| ✓ | | ✓ | **36.29** | 20.41 | 13.69 | 18.30 | 22.17 |
| ✓ | ✓ | ✓ | 32.72 | **21.01** | **16.45** | 22.25 | **23.11** |

Table 4: Effect of step $k$ (for updating the historical model) on the performance of DHA. We only show the results (R@mean) of the final model (trained across all four tasks) on each task under the CITM setting.

| $k$ | $T_1$ | $T_2$ | $T_3$ | $T_4$ | Avg. |
|---|---|---|---|---|---|
| 1 | 22.62 | 15.21 | 14.20 | **22.39** | 18.60 |
| 3 | 28.01 | 17.69 | 14.45 | 22.22 | 20.59 |
| 5 | 32.72 | 21.01 | **16.45** | 22.25 | **23.11** |
| 7 | **33.01** | **21.28** | 16.23 | 21.69 | 23.01 |

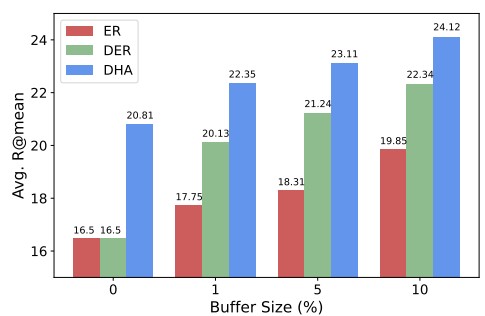

Figure 5: Results of DHA with extra memory buffer.

$T_t$ suffers from much less degradation (caused by 'Adapt with Hist') on task $T_t$. In other words, adopting both of the two strategies can ensure a good trade-off between the previous tasks and the current task during sequential training. (4) The extra memory buffer yields improvements in most cases. Moreover, even if the memory buffer is used, the two strategies of our DHA are still effective. This means that our DHA is complementary to the rehearsal-based methods.

We further conduct experiments to explore the effect of step $k$ on the performance of our DHA. Intuitively, if $k$ is too small, the historical model would be updated too frequently with the parameters of the main model. Therefore, although the knowledge gap is too small to affect the performance of the main model on the current task, the historical model is hard to preserve the historical knowledge. On the contrary, if $k$ is too large, the historical model would only have few updates with the parameters of the main model. As a result, a huge knowledge gap between the historical and main models would harm the performance of the main model on the current task. Overall, a good trade-off can be ensured by selecting the best $k$. Indeed, this analysis is validated by the results in Table 4. Specifically, the performance of our DHA on task $T_4$ is the best when $k = 1$ and gradually decreases when $k$ increases from 1 to 7, while the performance on the previous tasks $T_1$-$T_3$ grows higher at the same time. We thus select $k = 5$ with the highest average R@mean in this paper.

Finally, to investigate the effect of the buffer size, we make comparison among ER (Chaudhry et al., 2019), DER (Buzzega et al., 2020), and our DHA with different buffer sizes (0%, 1%, 5%, and 10% of the training data). We show the comparative results (average R@mean) in Figure 5. It can be seen that DHA beats ER and DER in all cases. Furthermore, our DHA with 1% buffer and 0% buffer even perform better than DER and ER with up to 10% buffer, respectively. This validates the effectiveness of direct parameter transfer (used in DHA) in continual learning.

## 5 CONCLUSION

In this paper, we propose a continual image-text modeling (CITM) setting, under which the model is required to be trained sequentially on four diverse image-text datasets and finally evaluated on all previous datasets. This new continual setting has a realistic application in large-scale image-text pre-training. We devise an effective dynamic historical adaptation (DHA) approach to coping with the forgetting problem in CITM. Different from existing continual learning methods, our DHA proposes to preserve the historical knowledge with direct parameter interaction between the historical and main models. Extensive experiments show the effectiveness of our DHA under the CITM setting.

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

## A    FULL ALGORITHM FOR DHA

In Algorithm 1, we give the pseudocode of the full algorithm for DHA. In Algorithm 2, we show our detailed strategy of constructing and updating memory buffer at the beginning of the current task. The core idea of Algorithm 2 is similar to ER-ring (Chaudhry et al., 2019). However, since ER-ring equally allocates the buffer to each class but the concept of 'class' does not exist in the image-text retrieval task, we keep the buffer evenly contains image-text pairs from all the previous datasets instead. The update strategy in Algorithm 2 is applied to all the competitors using memory buffer.

---

**Algorithm 1** Sequential Training with DHA

---

**Input:**   the dataset for sequential tasks $\{D_i\}_{i=1}^T$
     the main model with parameters $\theta_{main}$
     the historical model with parameters $\theta_{hist}$
     the best model of the last task with parameters $\theta_{last}$
     max iterations $i_{max}$ in each task
     hyperparameters $k, \lambda_1, \lambda_2$
**Output:**   the learned $\theta_{main}^*$
  initialize $\theta_{main}$ by training the main model on $D_1$
  initialize $\theta_{last} \leftarrow \theta_{main}$
  **for** $D_t \in \{D_2, ..., D_T\}$ **do**
     initialize $\theta_{hist} \leftarrow \theta_{last}, \theta_{main} \leftarrow \theta_{last}$          ▷ Initialize the main and historical models
     **for** $i \leftarrow 1$ to $i_{max}$ **do**
       **if** $i\%k = 0$ **then**
         $\theta_{hist}^i \leftarrow \lambda_2 \theta_{hist}^{i-1} + (1-\lambda_2)\theta_{main}^{i-1}$      ▷ Update the historical model with main model
       **else**
         $\theta_{hist}^i \leftarrow \theta_{hist}^{i-1}$             ▷ Do not update the historical model
       **end if**
       update $\theta_{main}^i$ according to Eq. (9)
     **end for**
     obtain the best $\theta_{main}$
     $\theta_{last} \leftarrow \theta_{main}$
  **end for**
  **return** the found best $\theta_{main}^*$

---

**Algorithm 2** Memory Buffer Update Strategy

---

**Input:**   the dataset $D_{t-1}$ of task $T_{t-1}$
     the memory buffer $M_{t-1}$ used in task $T_{t-1}$
     the memory buffer $M_t$ used in task $T_t$
     the samples $S_i^t$ selected from $D_i$ ($1 \leq i \leq t$) to compose buffer $M_t$
     the buffer size $|M|$
     the number of tasks $T$
**Output:**   $M_t$
  **if** $t = 2$ **then**
     $S_1^2 \xleftarrow{|M|} D_1$                ▷ Randomly select $|M|$ samples from $D_1$
     $M_2 \leftarrow S_1^2$
  **else if** $t \leq T$ **then**
     **for** $1 \leq i \leq (t-2)$ **do**
       $S_i^t \xleftarrow{\frac{|M|}{t-1}} S_i^{t-1}$            ▷ Randomly select $\frac{|M|}{t-1}$ samples from buffer $M_{t-1}$
     **end for**
     $S_{t-1}^t \xleftarrow{\frac{|M|}{t-1}} D_{t-1}$            ▷ Randomly select $\frac{|M|}{t-1}$ samples from task $T_{t-1}$
     $M_t \leftarrow \{S_1^t, S_2^t, ..., S_{t-1}^t\}$             ▷ Form the buffer $M_t$ used in task $T_t$
  **end if**
  **return** $M_t$

---

## B    IMPLEMENTATION DETAILS OF ALL COMPETITORS

We introduce the implementation details of all competitors for continual image-text modeling:

**LwF (Li & Hoiem, 2017)**: We adopt the best main model in the last task as the old model, maintain the old model in the new task, and align the output logits between the old and current main models as the distillation regularization.

**ER (Chaudhry et al., 2019)**: As the classic data-rehearsal approach, ER mainly reduces the catastrophic forgetting by reusing partial old data (from previous tasks) in the new task. We adopt Algorithm 2 (similar to ER-ring) to update the memory buffer. We set the buffer size as 5% of the training set of each task. Importantly, all the other competitors which adopt memory buffer share exactly the same buffer setting as ER.

**DER (Buzzega et al., 2020)**: We save logits instead of raw inputs and old models for DER. Following (Buzzega et al., 2020), we randomly select the previous model from 15 epochs to acquire various expressions of the buffer. Moreover, only the logits of previous sample pairs are aligned.

**CO2L (Cha et al., 2021)**: The data rehearsal and regularization strategies are both adopted by CO2L. Different from DER, CO2L applies data augmentation. Hence, we adopt two augmentations for input images in each mini-batch. Following its strategy, the sample pairs in the buffer are only used as negative samples.

We adopt the same temperature hyperparameter and learning rate for DHA, Baseline and all the above competitors. In this work, our DHA only adopts the same memory buffer as ER and mixes it with new data when training on a new task.

## C    DOMAIN SHIFT IN THE CITM SETTING

To directly demonstrate the domain shift across the four datasets used in our CITM setting, we conduct cross-dataset evaluation experiments. Concretely, we train the model (with the same architecture described in Sec. 3.2) independently on the train set of each dataset, and then evaluate it on the test sets of all four datasets to show its performance on the seen dataset and the other unseen datasets. As shown in Figure 6, the model achieves the highest performance on its seen dataset and much lower performance on unseen datasets. Therefore, we validate that there do exist domain shift, in other words, the domain gap across the four datasets. Overall, our CITM setting reasonably mimics the realistic application scenarios of image-text modeling.

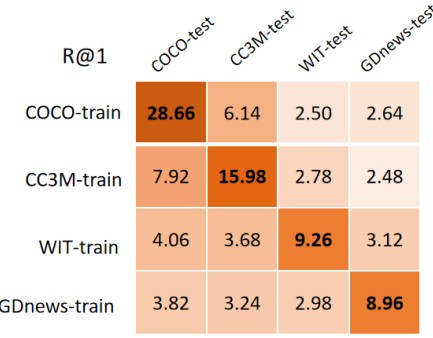

Figure 6: Results of cross-dataset evaluation after independent training.

## D    DETAILED EVALUATION RESULTS

More detailed experiments results are included in this section. Firstly, we show the ablative I2T results (Average R@mean) of our DHA model with different values of $\lambda_1$ and $\lambda_2$ in Table 5. According to their definitions in Sec. 3.3, $\lambda_1$ controls the update speed of the main model with the historical model, and $\lambda_2$ controls the update speed of the historical model with the main model. Thus, the balance should be made between the two coefficients. We can see that the best performance could be obtained when $\lambda_1 = 0.995$ and $\lambda_2 = 0.985$. Moreover, most of the combination groups in Table 5 lead to better results than DER (Buzzega et al., 2020) (21.24), which further indicates that our DHA is indeed a promising approach to CITM.

Table 5: Effect of coefficients $\lambda_1$ and $\lambda_2$ on the performance of DHA.

| $\lambda_1$ \ $\lambda_2$ | 0.98 | 0.985 | 0.99 | 0.999 |
|---|---|---|---|---|
| 0.98 | 20.68 | 21.01 | 20.93 | 20.87 |
| 0.99 | 21.24 | 22.23 | 21.64 | 21.33 |
| 0.995 | 22.15 | **23.11** | 22.28 | 22.12 |
| 0.999 | 21.43 | 22.84 | 21.97 | 22.08 |

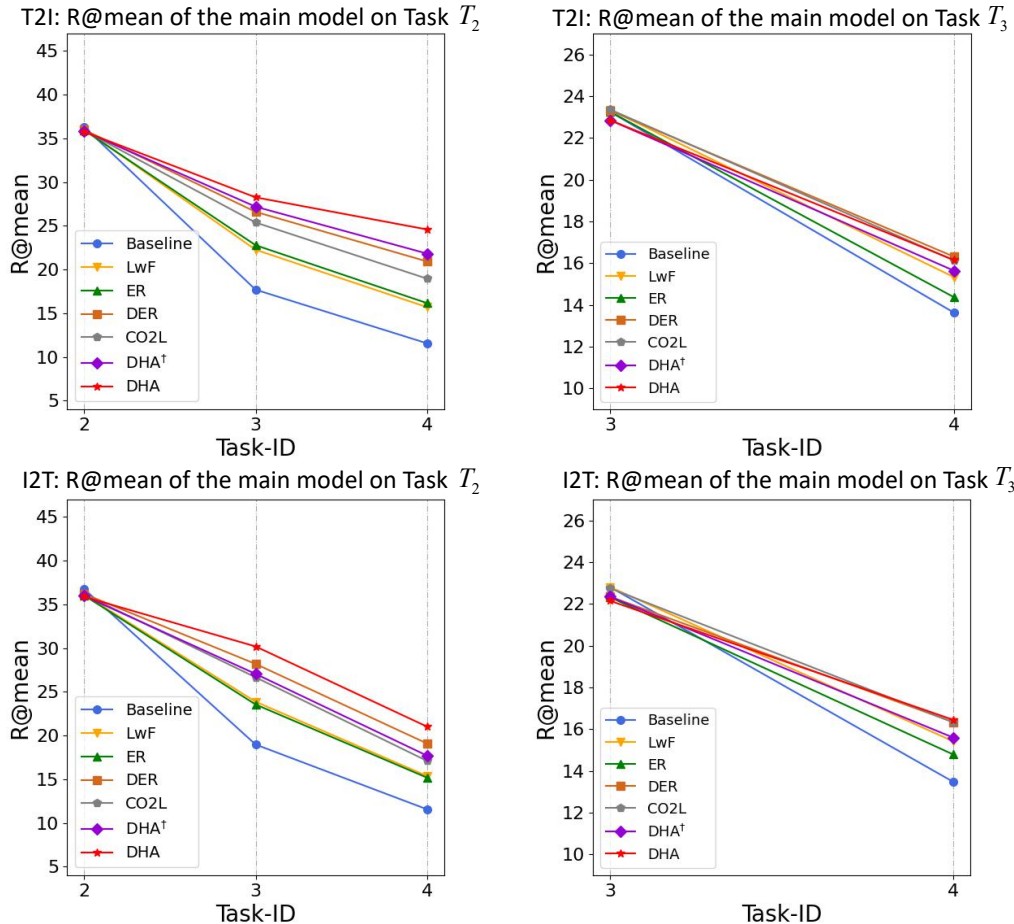

Figure 7: Illustration of the results of the main model (of all methods) on Task $T_2$ and Task $T_3$ during sequential training on the four tasks. 'T2I' denotes text-to-image retrieval and 'I2T' denotes image-to-text retrieval. It can be clearly seen that our DHA forgets with the lowest speed.

Secondly, we present the results on tasks $T_2$ and $T_3$ obtained by the main model during sequential training in Figure 7. It can be clearly seen that our DHA still forgets with the lowest speed on tasks $T_2$ and $T_3$, similar to Sec. 4.3 (see Figure 4). Importantly, by reviewing the results on tasks $T_1 - T_3$ (from Figure 4 and Figure 7), we find that DHA could address the catastrophic forgetting problem much better on the earlier seen tasks, which is a crucial advantage for practical applications.

Finally, we provide the detailed results in terms of Recall@1, Recall@5, and Recall@10), in addition to Table 1. We can observe from Tables 6–8 that our DHA consistently has the best performance on average recall and forgetting rate. Additionally, our DHA performs the best on all the three historical tasks (i.e., $T_1 - T_3$) in terms of Recall@1, which indicates that our DHA is indeed the best approach to CITM even with the most strict metric.

# E    FURTHER RESULTS FOR ABLATION STUDY

We compare the results between DHA-best-val (i.e., initializing the historical model with the best-validated model of the last task) and DHA-last-iteration (i.e., initializing the historical model with the last iteration model of the last task) under our framework in Table 9. It can be observed that these two strategies achieve almost the same performance.

Furthermore, the comparative results (R@mean) between training with all data (the upper bound) and training with DHA are presented in Table 10. We can find that there exists 5-6% gaps between our DHA and the upper bound in terms of the average performance (over all tasks).

Table 6: Comparative results (Recall@1 and FR) between our DHA and other representative/latest methods. 'T2I' denotes text-to-image retrieval and 'I2T' denotes image-to-text retrieval. All methods adopt the same network architecture. 'Mem' denotes the data rehearsal with 5% buffer.

| | Method | Mem | Task $T_1$ R@1 | FR | Task $T_2$ R@1 | FR | Task $T_3$ R@1 | FR | Task $T_4$ R@1 | Average R@1 | FR |
|---|---|---|---|---|---|---|---|---|---|---|---|
| T2I | Baseline | N | 4.61 | 77.73 | 4.92 | 72.82 | 4.62 | 48.09 | 9.72 | 5.97 | 66.21 |
| | LwF (Li & Hoiem, 2017) | N | 6.04 | 70.82 | 6.00 | 66.85 | 5.22 | 43.38 | **9.82** | 6.77 | 60.35 |
| | ER (Chaudhry et al., 2019) | Y | 5.62 | 72.85 | 5.78 | 67.34 | 4.78 | 45.06 | 9.34 | 6.38 | 61.75 |
| | DER (Buzzega et al., 2020) | Y | 7.65 | 63.04 | 8.24 | 53.91 | 5.64 | 35.17 | 8.68 | 7.55 | 50.71 |
| | CO2L (Cha et al., 2021) | Y | 7.30 | 64.73 | 6.90 | 59.36 | 5.74 | 37.33 | 9.20 | 7.29 | 53.81 |
| | DHA† (ours) | N | 8.03 | 61.21 | 8.74 | 49.89 | 5.73 | 35.98 | 8.73 | 7.81 | 49.03 |
| | DHA (ours) | Y | **9.31** | **55.02** | **9.30** | **43.77** | **5.98** | **25.44** | 8.46 | **8.26** | **41.41** |
| I2T | Baseline | N | 5.88 | 79.48 | 3.88 | 78.77 | 4.72 | 47.20 | 9.74 | 6.06 | 68.48 |
| | LwF (Li & Hoiem, 2017) | N | 8.64 | 69.85 | 5.42 | 70.29 | 5.50 | 44.11 | **10.04** | 7.40 | 61.42 |
| | ER (Chaudhry et al., 2019) | Y | 7.52 | 73.76 | 5.20 | 70.15 | 5.24 | 45.30 | 9.12 | 6.77 | 63.07 |
| | DER (Buzzega et al., 2020) | Y | 11.40 | 60.22 | 6.98 | 60.83 | 6.02 | 38.07 | 9.20 | 8.40 | 53.04 |
| | CO2L (Cha et al., 2021) | Y | 10.68 | 62.74 | 6.34 | 62.84 | 6.22 | 34.80 | 9.70 | 8.24 | 53.46 |
| | DHA† (ours) | N | 11.42 | 60.15 | 6.60 | 59.33 | 5.56 | 36.12 | 8.74 | 8.08 | 51.86 |
| | DHA (ours) | Y | **14.52** | **49.34** | **8.32** | **48.77** | **6.22** | **34.86** | 9.12 | **9.55** | **44.32** |

Table 7: Comparative results (Recall@5 and FR) between our DHA and other representative/latest methods. 'T2I' denotes text-to-image retrieval and 'I2T' denotes image-to-text retrieval. All methods adopt the same network architecture. 'Mem' denotes the data rehearsal with 5% buffer.

| | Method | Mem | Task $T_1$ R@5 | FR | Task $T_2$ R@5 | FR | Task $T_3$ R@5 | FR | Task $T_4$ R@5 | Average R@5 | FR |
|---|---|---|---|---|---|---|---|---|---|---|---|
| T2I | Baseline | N | 14.36 | 69.62 | 13.62 | 65.62 | 14.32 | 41.84 | 24.38 | 16.67 | 59.03 |
| | LwF (Li & Hoiem, 2017) | N | 17.88 | 62.17 | 16.88 | 56.54 | 16.32 | 35.49 | 24.20 | 18.82 | 51.40 |
| | ER (Chaudhry et al., 2019) | Y | 17.27 | 63.47 | 17.66 | 54.88 | 15.44 | 37.18 | 23.44 | 18.45 | 51.84 |
| | DER (Buzzega et al., 2020) | Y | 21.93 | 53.61 | 22.80 | 41.93 | **17.52** | 27.36 | 22.86 | 21.28 | 41.71 |
| | CO2L (Cha et al., 2021) | Y | 20.87 | 55.85 | 20.22 | 47.78 | 17.60 | 30.32 | **25.04** | 20.93 | 44.65 |
| | DHA† (ours) | N | 23.11 | 51.11 | 23.67 | 39.03 | 16.48 | 30.38 | 22.92 | 21.55 | 40.17 |
| | DHA (ours) | Y | **26.65** | **43.62** | **24.78** | **34.65** | 16.80 | 30.12 | 22.50 | **22.68** | **36.13** |
| I2T | Baseline | N | 18.40 | 67.22 | 12.12 | 70.19 | 14.32 | 44.41 | **27.47** | 18.08 | 60.61 |
| | LwF (Li & Hoiem, 2017) | N | 23.14 | 58.78 | 16.18 | 59.97 | 16.56 | 37.32 | 25.04 | 20.23 | 52.02 |
| | ER (Chaudhry et al., 2019) | Y | 23.08 | 58.89 | 16.60 | 58.52 | 15.92 | 38.58 | 24.14 | 19.94 | 52.00 |
| | DER (Buzzega et al., 2020) | Y | 30.42 | 46.63 | 19.64 | 48.71 | 16.54 | 31.25 | 24.06 | 22.67 | 42.20 |
| | CO2L (Cha et al., 2021) | Y | 28.08 | 49.98 | 18.44 | 53.32 | 17.70 | 32.90 | 25.68 | 22.48 | 45.40 |
| | DHA† (ours) | N | 30.36 | 45.39 | 19.22 | 49.88 | 15.72 | 32.47 | 24.40 | 22.43 | 42.58 |
| | DHA (ours) | Y | **35.24** | **37.23** | **22.76** | **40.67** | **17.84** | **28.66** | 24.02 | **24.97** | **35.52** |

Table 8: Comparative results (Recall@10 and FR) between our DHA and other representative/latest methods. 'T2I' denotes text-to-image retrieval and 'I2T' denotes image-to-text retrieval. All methods adopt the same network architecture. 'Mem' denotes the data rehearsal with 5% buffer.

| | Method | Mem | Task $T_1$ R@10 | FR | Task $T_2$ R@10 | FR | Task $T_3$ R@10 | FR | Task $T_4$ R@10 | Average R@10 | FR |
|---|---|---|---|---|---|---|---|---|---|---|---|
| T2I | Baseline | N | 21.96 | 64.06 | 19.80 | 61.37 | 21.96 | 37.11 | 33.76 | 24.37 | 54.18 |
| | LwF (Li & Hoiem, 2017) | N | 26.50 | 56.64 | 24.20 | 51.81 | 24.46 | 30.47 | 34.02 | 27.30 | 46.31 |
| | ER (Chaudhry et al., 2019) | Y | 26.00 | 57.45 | 25.02 | 50.94 | 22.90 | 33.89 | 31.94 | 26.47 | 47.43 |
| | DER (Buzzega et al., 2020) | Y | 31.98 | 47.67 | 31.72 | 37.31 | **25.76** | 24.77 | 31.58 | 30.26 | 36.58 |
| | CO2L (Cha et al., 2021) | Y | 30.74 | 50.52 | 29.72 | 40.39 | 25.06 | 29.49 | **34.60** | 30.03 | 40.13 |
| | DHA† (ours) | N | 32.80 | 46.33 | 33.06 | 34.51 | 24.71 | 28.06 | 32.46 | 30.76 | 36.30 |
| | DHA (ours) | Y | **37.78** | **38.18** | **34.76** | **29.89** | 25.26 | 25.02 | 32.70 | **32.63** | **31.03** |
| I2T | Baseline | N | 27.50 | 60.52 | 18.64 | 63.75 | 21.40 | 38.93 | 33.94 | 25.37 | 51.77 |
| | LwF (Li & Hoiem, 2017) | N | 33.00 | 52.63 | 24.48 | 51.83 | 24.20 | 34.24 | 34.80 | 29.12 | 46.23 |
| | ER (Chaudhry et al., 2019) | Y | 33.10 | 52.48 | 23.72 | 52.92 | 23.20 | 35.34 | 32.84 | 28.22 | 46.91 |
| | DER (Buzzega et al., 2020) | Y | 41.30 | 40.71 | 29.60 | 41.41 | 25.34 | **28.54** | 32.42 | 32.14 | 36.89 |
| | CO2L (Cha et al., 2021) | Y | 39.94 | 42.66 | 26.50 | 47.69 | 25.06 | 29.92 | **35.38** | 31.72 | 40.09 |
| | DHA† (ours) | N | 41.96 | 39.76 | 27.26 | 45.01 | 22.72 | 31.15 | 34.16 | 31.53 | 38.64 |
| | DHA (ours) | Y | **48.40** | **30.52** | **31.96** | **35.54** | **25.30** | 29.35 | 33.62 | **34.82** | **31.80** |

Table 9: Comparative results between DHA with the best main model (of the last task) as the historical model and DHA with the last-iteration model as the historical model. 'T2I' denotes text-to-image retrieval and 'I2T' denotes image-to-text retrieval.

| | Method | Task $T_1$ | | Task $T_2$ | | Task $T_3$ | | Task $T_4$ | Average | |
|---|---|---|---|---|---|---|---|---|---|---|
| | | R@mean | FR | R@mean | FR | R@mean | FR | R@mean | R@mean | FR |
| T2I | DHA-best-val | 24.58 | 42.88 | 22.95 | 33.82 | 16.15 | 24.50 | 21.22 | 21.29 | 33.73 |
| | DHA-last-iteration | 24.47 | 43.13 | 23.06 | 33.51 | 16.01 | 25.15 | 21.40 | 21.24 | 33.93 |
| I2T | DHA-best-val | 32.72 | 36.45 | 21.01 | 39.50 | 16.45 | 24.71 | 22.25 | 23.11 | 33.55 |
| | DHA-last-iteration | 32.67 | 36.56 | 21.22 | 38.91 | 16.21 | 25.81 | 22.53 | 23.16 | 33.76 |

Table 10: Comparative results (R@mean) between training with all data (upper bound) and training with DHA. 'T2I' denotes text-to-image retrieval and 'I2T' denotes image-to-text retrieval.

| Method | T2I | | | | | I2T | | | | |
|---|---|---|---|---|---|---|---|---|---|---|
| | Task $T_1$ | Task $T_2$ | Task $T_3$ | Task $T_4$ | Average | Task $T_1$ | Task $T_2$ | Task $T_3$ | Task $T_4$ | Average |
| Upper bound | 31.61 | 32.39 | 18.86 | 22.69 | 26.39 | 41.89 | 32.27 | 20.82 | 22.38 | 29.34 |
| DHA (ours) | 24.58 | 22.95 | 16.15 | 21.22 | 21.29 | 32.72 | 21.01 | 16.45 | 22.25 | 23.11 |

Table 11: Evaluation results for setting different values of $\lambda_1$ for the text and image modalities. 'T2I' denotes text-to-image retrieval and 'I2T' denotes image-to-text retrieval.

| | Setting 1 | Setting 2 | Setting 3 | Setting 4 |
|---|---|---|---|---|
| $\lambda_1$ for the text modality | 0.993 | 0.993 | 0.995 | 0.995 |
| $\lambda_1$ for the image modality | 0.993 | 0.995 | 0.993 | 0.995 |
| T2I: average R@mean | 20.77 | 21.16 | **21.45** | 21.29 |
| I2T: average R@mean | 21.43 | 22.94 | **23.33** | 23.11 |

Additionally, we clarify that applying the same value of $\lambda_1$ for image and text modalities enables the proposed DHA to be easily deployed in other continual learning settings. However, we should point out that our proposed framework also provides convenience for discovering and making use of modality-specific forgetting characteristics under the CITM setting. Due to the generality (flexibility) of the proposed DHA, we can easily utilize it to cope with modality-specific forgetting (or modality-wise domain gap) by setting different values of $\lambda_1$ (in Eq. (9)) for the text and image modalities. The obtained results are shown in Table 11. We can see that the proposed DHA leads to further improvements when $\lambda_1$ takes different values for the two modalities. This also suggests that coping with modality-specific forgetting is indeed necessary for our CITM setting.

