# OpenReview forum: "Dynamic Historical Adaptation for Continual Image-Text Modeling"
_ICLR.cc/2023/Conference — Submitted to ICLR 2023_

### Official Review · Reviewer_4eZi · 2022-10-24

**Confidence:** 4
**Correctness:** 4
**Technical Novelty And Significance:** 4
**Empirical Novelty And Significance:** 4
**Recommendation:** 8

**Clarity, Quality, Novelty And Reproducibility:**

The paper is clear, and the proposed method seems to be novel. The method seems to be replicable from the details in the paper.

**Strength And Weaknesses:**

Strengths:
+ New formulation of continual learning with minimal forgetting
+ Thorough evaluation and ablations
+ Well written paper


Questions:
- Performance on Task3 and Task4.
There seems to be a weakness of the propose model that shows in Table 1 for tasks Tasks 3 and 4. Do they authors have any analysis showing why this is happening. I would assume the performance should be best in the later Tasks than the earlier tasks.

**Summary Of The Paper:**

This paper proposes a method for continual learning in text-image modeling. They propose a formulation that uses a historical and main neural networks whose parameters are used interchangeably	between the two networks during parameter optimization as a weighed average. In experiments, the authors compare favorably against baseline methods through thorough evaluation and they also provide a thorough ablation to highlight the contribution of the different components and hyper parameters of the proposed method.

**Summary Of The Review:**

The formulation in this paper is interesting, and the authors made sure to ablate each portion of it to show the different behaviors with different pieces. In addition, the authors outperform the baselines on average. Therefore, I am leaning towards accepting this paper.

---

### Official Review · Reviewer_xMpX · 2022-10-24

**Confidence:** 3
**Correctness:** 3
**Technical Novelty And Significance:** 2
**Empirical Novelty And Significance:** 2
**Recommendation:** 5

**Clarity, Quality, Novelty And Reproducibility:**

Clarity: good, the proposed method is simple.

Quality: limited, since I think the overall benchmark and pipeline do not seem to address the key issues of incremental learning.

Novelty: hard to judge -- if the overall design is fine and the direction is ok, I will say the novelty is sufficient.

Reproducibility: seems good, the method is very simple anyway.

**Details Of Ethics Concerns:**

Not writing ethical concerns in the paper.

**Strength And Weaknesses:**

Strengths
1. The studied problem is interesting and important.
2. The paper is easy to follow.

Weaknesses

I am familiar with VL understanding but I do not work in the continual learning field (I do know too much about it). I choose to write down my opinions below and wait for other reviewers' comments and discussions with the authors.

I think the proposed method is reasonable but it does not provide new insights. It is about propagating information back and forth between an online (new) model and an offline (old) model. Intuitively, by absorbing knowledge from the old model, the recognition accuracy on old datasets is expected to grow, but that on new data is expected to drop. Experiments validate the hypothesis, where we see a relatively large accuracy gain on task T1, but the gain drops rapidly and eventually becomes a deficit on task T4. Although the average accuracy is higher than the competitors, I guess it comes from a better balance among the tasks -- that said, the core issue of catastrophic forgetting is not well addressed, but the approach seems to add weights among different, historical models toward a better average accuracy. Personally, I do not praise the contribution of the proposed method.

I shall admit that I did not achieve a clear thought of evaluating this paper. It seems that replaying old data (with buffer) and average old and new models are the only way to alleviate forgetting, which is trivial to me. Is this the correct direction for incremental learning?

There are some minor issues, such as the missing of diagnosis of the key hyper-parameter, k.

**Summary Of The Paper:**

This paper presents an incremental learning approach for VL understanding (in particular, cross-modal retrieval). The authors established a benchmark by combining several popular VL datasets and defining an incremental (sequential) scenario. The proposed method involves applying EMA back and forth between old (offline) and new (online) models. Experiments show improvement in the studied scenario.

**Summary Of The Review:**

I had a hard time evaluating the contribution of this paper. Temporarily, I do not think the paper made important contributions that shall be published at ICLR.

---

> ### Author Response · Authors · 2022-11-17
> **Look forward to author-reviewer discussion**
>
> Dear Reviewer xMpX,
>
> Thanks again for your insightful suggestions and comments. In our rebuttal, we have made pointwise responses to your comments. As the deadline of author-reviewer discussion is approaching, we are happy to provide any additional clarifications or experiments that you may need.
>
> Thank you for your time!
>
> Best, \
> Authors

---

### Official Review · Reviewer_4FxS · 2022-10-25

**Confidence:** 3
**Correctness:** 4
**Technical Novelty And Significance:** 3
**Empirical Novelty And Significance:** 3
**Recommendation:** 8

**Clarity, Quality, Novelty And Reproducibility:**

The paper is clear and we can easily follow it. Basically, the proposed model is novel to me and I think we can reproduce it.

**Strength And Weaknesses:**

Strength:

S1: This paper identifies the important role of direct parameter transfer (between the historical and main models) in continual learning, and proposed a dynamic historical adaptation model.

S2: The proposed model outperforms its counterparts, including rehearsal-based methods, regularization-based methods and hybrid models.

S3: Extensive experiments are conducted to show the effectiveness of the proposed model.

Weaknesses:

W1: It is unclear how large the gap is between using all data to train a model and the proposed DHA.

W2: I am wondering whether the order of tasks matters.

W3: The style of Flickr30k is similar to MSCOCO, and there should be overlaps with MSCOCO, so zero-shot may not be appropriate.

**Summary Of The Paper:**

This paper focuses on continual image-text embedding, reducing the computation and storage resources using all data in the training phase. For the first time, the paper identifies the important role of direct parameter transfer (between the historical and main models) in continual learning and the proposed DHA outperforms existing continual learning approaches, including rehearsal-based methods, regularization-based methods and hybrid models.

**Summary Of The Review:**

The paper focuses on continual image-text embedding, which is an important task. The proposed DHA model outperforms its counterparts, reducing the computation and storage resources of using all data to train models. In this phase, I give it a score of 8.

---

### Official Review · Reviewer_miUb · 2022-10-25

**Confidence:** 4
**Correctness:** 3
**Technical Novelty And Significance:** 2
**Empirical Novelty And Significance:** 3
**Recommendation:** 5

**Clarity, Quality, Novelty And Reproducibility:**

- Clarity : The paper is well-written and easy to follow.
- Quality : The presentation of the paper is neat.
- Novelty : The proposed problem setting (CITM) has not been addressed so far, but is a relevant and novel problem to ML community.
- Reproducibility : The authors state that they will release the datasets and code soon in the Section 4.2 (Implementation Details).

**Strength And Weaknesses:**

- Strengths
    - The proposed novel CITM problem has not been studied, but is a real use case, since many image-text datasets with large domain shifts have been proposed recently, and vision-language models are expensive to retrain in general.
    - The proposed framework demonstrates better performance than other baselines for the newly constructed benchmarks, using existing image-text datasets.
    - The idea of alternatively updating the historical model and the main model seems interesting, but the reviewer is not fully convinced yet.
    - The ablation study well supports the claim that “Adapt with Hist” provides stability while “Dynamic Hist” addresses plasticity.

- Weaknesses
    - It seems the proposed framework is not fully aligned with the proposed CITM setting.
        - First, the reviewer agrees with the authors’ claim that the image-text datasets have a large domain gap, and thus the existing rehearsal-based and regularization-based methods may not be suitable to the CITM setting.
        - Therefore, the main problem of the CITM setting (according to the paper) is domain shift.
        - However, the domain shift problem arises from collecting multiple datasets, which is not incurred from the inheritance of multi-modal data.
        - Thus the link between the proposed problem setting and the proposed framework is weak.
        - For me, this kind of study should deliver some idea or solution which handles multi-modal learning via multi-modality itself or modality-specific aspects. There could be modality-specific forgetting, or modality-wise domain gap, etc.,.
        - What makes the proposed framework special for the multi-modal task, not just for the datasets with large domain shifts?
    - Discussion for the methodology
        - Finding the best model for the last task using the test seems inappropriate. Although there could be some works beyond my knowledge using a test set of the previous task, continual learning usually assumes that we do not have access to the test set during incremental training steps.
        - A more common way to choose the model of the last task is to choose the model after the last iteration.
        - What if we just select the model after the last iteration for each task? Or, what is the mean and variance of epoch (or iteration) for each task under the current version of the framework?
    - Discussion for the datasets
        - It seems subsampling the same number of data from each dataset (e.g., $T_2-T_4$) diminishes the necessity of CITM setting, while this strategy addresses the dataset-imbalance problem for the experiments. In general, different image-text datasets have different magnitudes of sizes. The obvious examples are MSCOCO (123K images) and CC3M (3M images), which are used in this study.
        - The order of the datasets is the crucial aspect that affects the performance of continual learning. The results for different dataset orders could make the authors’ claim more robust. It could be a minor point if we sample the same number of data from each dataset as in this study. However, it would be crucial if we just naively use all data in the datasets with different magnitudes of size.
    - Minor
        - Typo : wrok → work : last sentence in Appendix B.

**Summary Of The Paper:**

This paper addresses the problem of continual learning under a novel Continual Image-Text Modeling setting where the model learns a sequence of multiple image-text matching tasks. The authors claim that transferring the old knowledge from the historical model parameters and periodically updating the current model to the historical model addresses the problem, since the datasets used in the proposed CITM setting have a large domain gap. The experimental results on the proposed benchmarks show the effectiveness of the proposed framework compared to the baselines.

**Summary Of The Review:**

Although the paper proposes a novel and interesting problem setting, the reviewer has some concerns to be addressed, as stated in the weakness and questions section.

---

> ### Author Response · Authors · 2022-11-17
> **Look forward to author-reviewer discussion**
>
> Dear Reviewer miUb,
>
> Thanks again for your insightful suggestions and comments.  In our rebuttal, we have made pointwise responses to your comments. We have also added new experiment results to answer Q1 and Q2.
>
> As the deadline of author-reviewer discussion is approaching, we are happy to provide any additional clarifications or experiments that you may need.
>
> Thank you for your time!
>
> Best,\
> Authors

---

### Decision · Program_Chairs · 2023-01-20

**Decision:**

Reject

**Justification For Why Not Higher Score:**

This paper might have been numerically a borderline paper (well, on the borderline of borderline), but the two very positive reviews were neither detailed nor particularly insightful. The two reviewers with more negative reviews actively engaged with both authors and AC to hash out a consensus opinion that the paper has defects that outweigh its positive aspects.

**Justification For Why Not Lower Score:**

N/A

**Metareview: Summary, Strengths And Weaknesses:**

# Summary of Contribution
This paper describes an approach to continual vision/language modeling for cross-modal (image <--> text) retrieval problems. The authors propose to transfer knowledge from "historical" model parameters and dynamically update the current model. This technique, which the authors term Dynamic Historical Adaptation (DHA) in which weights of the best previous model are used as a starting point for learning the new task. The authors propose an incremental learning scenario considering four cross-modal retrieval tasks (MS-COCO, CC3M, WIT, and GoodNews).

# Strengths
* **New CL Benchmark:** The main technical contribution of the paper (DHA) is a somewhat naive approach to balancing between old and new models -- something that is not particularly novel, and the authors do not motivate why it is specifically appropriate to the vision/language modeling task at hand. This is seen, as observed by one reviewer, in how this balancing yields the most impressive improvement on early tasks seems compared to relatively little (or even no) gains on later ones. As such, it represents a rather marginal contribution to the theory and practice of continual learning, and as such is less appropriate for a top ML venue like ICLR.
* **Experimental Protocol:** Two reviewers raise significant concerns about the appropriateness of the splits (and their order) used in the experimental protocol for incremental learning of cross-modal retrieval models. The authors argue that their method allows for more realistic continual learning of Image-Text models, however the subsampling of datasets is inherently unrealistic -- especially given the observation from one reviewer that large, domain-specific datasets typically become available after smaller, more general-purpose ones. Calibrating the size of datasets to the first (and using a single ordering) is a limit in the experimental evaluation.

# Weaknesses
* ** Marginal Contribution:**  The main technical contribution of the paper (DHA) is a somewhat naive approach to balancing between old and new models -- something that is not particularly novel, and the authors do not motivate why it is specifically appropriate to the vision/language modeling task at hand. This is seen, as observed by one reviewer, in how this balancing yields the most impressive improvement on early tasks seems compared to relatively little (or even no) gains on later ones. As such, it represents a rather marginal contribution to the theory and practice of continual learning, and as such is less appropriate for a top ML venue like ICLR.

* **Experimental Protocol:** Two reviewers raise significant concerns about the appropriateness of the splits (and their order) used in the experimental protocol for incremental learning of cross-modal retrieval models. The authors argue that their method allows for more realistic continual learning of Image-Text models, however the subsampling of datasets is inherently unrealistic -- especially given the observation from one reviewer that large, domain-specific datasets typically become available after smaller, more general-purpose ones. Calibrating the size of datasets to the first (and using a single ordering) is a limit in the experimental evaluation.

# Summary
There was a lively back-and-forth between some reviewers and authors during the discussion phase. In the end, the reviewers were unconvinced by the author rebuttals to their primary concerns, especially those related to the experimental protocol. While it is important to consider new and complex scenarios like cross-modal retrieval for continual learning, the contributions of the proposed approach to CITL theory and practice are very marginal (DHA, a balancing of old and new models from section 3.3). The paper and the proposed approach provide no insight into the CITL problem that is in fact *specific* to CITL, and moreover, multiple reviewers raised significant concerns about the experimental splitting protocols. For these reasons, the paper does not meet the bar for acceptance to ICLR as its contributions are marginal and of interest to only a small subset of the community. The area chair, senior area chair, and program chair reach this decision after reading this paper, reviews, rebuttals, and reviewer discussions.